

# Enhancing effect of 5-azacytidine on saline–alkaline resistance of *Akebia trifoliata* and underlying physiological and transcriptomic mechanisms

Xiao Xu Bi[*], Kai Wang[*], Xiaoqin Li, Jiao Chen, Jin Yang, Jin Yan, Guijiao Wang and Yongfu Zhang

School of Agriculture and Life Sciences, Kunming University, Kunming, Yunnan, China
[*] These authors contributed equally to this work.

Corresponding author
Yongfu Zhang, 123017360@qq.com

## ABSTRACT

Saline-alkaline stress is a common problem in *Akebia trifoliata* cultivation. In this study, the enhancing effects of 5-azacytidine (5-AzaC) on the resistance of *A. trifoliata* to saline–alkaline stress and the underlying mechanisms were investigated. Plant height, stem diameter, biomass, root length, fresh weight of root, and root/shoot ratio of 6-month-old *A. trifoliata* seedlings were measured after saline–alkaline stress with or without 5-AzaC treatment. Moreover, the contents of photosynthetic pigments, malondialdehyde (MDA), $H_2O_2$, sodium, soluble sugar, and proline; activities of superoxide dismutase, peroxidase (POD), and catalase (CAT); and anatomical structures of root, stem, and leaf were assessed. Furthermore, comparative transcriptome sequencing was performed. The results demonstrated that growth and development of *A. trifoliata* were severely inhibited under saline–alkaline stress, suggesting that the seedlings were exposed to severe oxidative and osmotic stresses. Treatment with exogenous 5-AzaC could significantly relieve the symptoms of saline–alkaline stress in *A. trifoliata*. Under saline–alkaline stress, 5-AzaC could increase the stem diameter, biomass, root length, fresh weight of root, and root/shoot ratio and minimize damages to the anatomical structure. Moreover, absorption of $Na^+$ was reduced; ionic balance was maintained; POD and CAT activities were significantly improved; proline and soluble sugar contents increased, and $H_2O_2$ and MDA contents decreased. Transcriptome analysis revealed that 5-AzaC functioned via regulating KEGG pathways such as plant hormone signal transduction, phenylpropanoid biosynthesis, photosynthesis, amino sugar and nucleotide sugar metabolism, and glutathione metabolism under saline–alkaline stress. Particularly, enhanced expression of genes from the auxin pathway in plant hormone signal transduction; the lignin synthetic pathway in phenylpropanoid biosynthesis; and photosystem II, photosystem I, photosynthetic electron transport, and F-type ATP pathway in photosynthesis may be related to 5-AzaC-induced saline–alkaline resistance. The results provided theoretical references for *A. trifoliata* cultivation in saline–alkaline soil and application of 5-AzaC to improve saline–alkaline tolerance in plants.

## INTRODUCTION

Soil salinization has become a global problem affecting agricultural production in India, Australia, China, the United States, and more than 100 countries and regions. Salinized soil accounts for approximately 1/4th of the global land area, and due to natural and anthropogenic factors, saline-alkaline land area is increasing year by year (*Gamalero et al., 2020*; *Zhang et al., 2022*). Soil salinization is a very serious issue in the coastal and northwest regions of China, directly affecting agricultural production and development of China (*Wu et al., 2019*). Under the high-salt environment, plants are forced to absorb a large number of salt ions. Excessive salt ions disrupt the dynamic balance of the reactive oxygen metabolic system; this damages membrane lipids or membrane proteins, increases membrane permeability and intracellular water solute extravasation, and promotes physiological drought (*Sadder et al., 2020*). The damage caused to plants by saline–alkaline stress includes osmotic effects due to salt stress, ion imbalance, water deficiency, and nutrient deficiency, ultimately leading to oxidative stress in plants. Additionally, saline-alkaline stress may cause high-pH stress due to the presence of alkaline salts (*Basu et al., 2020*). Under saline-alkaline stress, plant cells exhibit increased production of reactive oxygen species (ROS) in the form of free radicals or non-free radicals. Excessive ROS production leads to oxidative damage of cellular proteins, lipids, nucleic acids, and plasma membrane; significant decrease in the uptake of phosphorus and potassium; and increase in the uptake of toxic ions such as $Na^+$ and $Cl^-$, negatively affecting the growth and yield of crops (*Paheli & Debasis, 2021*; *Keyikoglu, Aksu & Arslan, 2021*).

Salt stress is very harmful to plant growth, and plants cannot escape from salt stress damage because their root system is fixed to the land. However, they have developed complex salt-resistance mechanisms in the long-term salt-stress environment. The common saline-alkaline resistance mechanisms include resistances to osmotic stress, oxidative stress, and ionic stress. First, under osmotic stress, plant roots accumulate small molecules, including proline, $Na^+$, $K^+$, $NO_3^-$, and soluble sugar, to regulate osmotic pressure in the plant and maintain the osmotic balance of cells and tissues (*Tang, Harris & Newton, 2003*). Further, under oxidative stress, activities of antioxidant enzymes such as superoxide dismutase (SOD), peroxidase (POD), and catalase (CAT) are enhanced, contributing to the removal of ROS (*Wang et al., 2024*). Additionally, when plants are subjected to saline-alkaline stress, intracellular $Na^+$ concentration decreases and $K^+$ concentration rapidly increases. As the extent of saline-alkaline stress increases, the $Na^+/K^+$ ratio in seedlings decreases. The presence of excessive $Na^+$ in plants leads to $K^+$ deficiency, inhibiting the uptake of other ions ($Ca^{2+}$ and $Mg^{2+}$). Effective control of $Na^+$-$K^+$ uptake and transport can improve saline–alkaline resistance (*AbdElgawad et al., 2016*; *Zhu et al., 2016*).

It has been demonstrated that 5-azacytidine (5-AzaC), as a methylation inhibitor, enables plants to acquire new traits and generate new functions. 5-AzaC treatment in *Brassica oleracea var. botrytis* L. (*Li et al., 2010*) and *Rhododendron simsii* Planch. (*Zhou et al., 2016a*) could reduce the DNA methylation level and promote early flowering. 5-AzaC treatment could slow down the growth of *B. rapa var. glabra* Regel seedlings under high-temperature stress, slow down the reduction in protein content and POD activity,

and reduce the malondialdehyde (MDA) content and cell membrane permeability (*Zhong et al., 2013*). Additionally, 5-AzaC treatment improved salt resistance in wheat (*Zhong, Xu & Wang, 2010*), *Arabidopsis thaliana (L.)* Heynh. (*Ogneva et al., 2019*), and *Hibiscus cannabinus* L. (*Li et al., 2021*) under salt stress. However, it is unclear whether 5-AzaC improves salt resistance in *Akebia trifoliata*.

*A. trifoliata* (Thunb.) Koidz., with important value as a medicine and food, exhibits advantages including high yield, drought resistance, extensive growth management, and easy transplantation and planting. Its fruits are unique in taste and rich in sugar, vitamin C, and 12 amino acids. They can be used as raw materials for brewing and preparations of health food, fruit tea, and preserved fruits. Therefore, *A. trifoliata* has high nutritional and economic value. *A. trifoliata* is susceptible to saline-alkaline stress, which decrease the contents of soluble sugar, chlorophyll, and carotenoid. This negatively affects the growth and development of *A. trifoliata* and taste of its fruit.

The role of 5-AzaC however in saline-alkali resistance in *A. trifoliate* has not been previously reported. In this study, the effect of 5-AzaC on the resistance of *A. trifoliata* to saline-alkaline stress was investigated firstly. The roots of *A. trifoliata* were subjected to saline-alkaline stress with or without 5-AzaC treatment, and the morphology, physiological and biochemical responses, anatomical structure, non-parametric transcriptome sequencing, and bioinformatic analysis of the sequencing results were performed to reveal the effect and mechanism of action of 5-AzaC. This study provided technical support for the optimal growth and development of *A. trifoliata* in saline-alkaline soil and references for the analysis of related functional genes and molecular breeding in future.

## MATERIALS AND METHODS
### Materials and sample preparation
For this study, 6-month-old cuttings of *A. trifoliata* seedlings were obtained from Qiubei County, Wenshan Prefecture, Yunnan Province. The experiment was conducted in the Agricultural Practice Park of Kunming College, Kunming, Yunnan Province. The process of cultivation was as follows. The lignified branches from winter pruning were cut. The morphological upper and lower ends were cut with flat and slant cuts, respectively. Overall, 2–3 axillary buds were retained in each branch. The morphological lower end was inserted into a seedling bag containing substrate (peat:vermiculite:perlite = 3:1:1). When the seedlings grew to 40–50 cm, healthy and uniformly grown seedlings free from pests and diseases were selected and planted in a bowl (diameter = 25 cm, height = 30 cm) with substrate (one plant per bowl). They were watered with 500 mL of Hoagland's solution (pH = 6.5) once a week. After 15 d, pests- and disease-free plants with uniform growth were selected and placed in white pots (100 cm × 35 cm × 35 cm) along with the contents of bowl. Four plants were placed in each pot. These pots were used for further treatments.

Three treatments were set up: Con (blank control), Salt (150 mmol/L Na$^+$), and Salt+5-AzaC (200 $\mu$mol/L 5-AzaC + 150 mmol/L Na$^+$) (*Koetle et al., 2023*). Each treatment involved three biological replicates. 5-AzaC (klamar; purity = 98%) was dissolved in sterile water to prepare 100 mmol/L solution, which was diluted 1,000-fold for use. Sodium salts

($Na_2SO_4$:$Na_2CO_3$:$NaHCO_3$ = 1:1:1) were dissolved in Hoagland's solution to prepare a solution with $Na^+$ concentration of 150 mmol/L (pH = 8.5). For the Salt and Salt+5-AzaC groups, 1 L of this salt solution was poured in the white pots once a week. For the Salt+5-AzaC group, at the same time, each plant was uniformly sprayed with 200 µmol/L 5-AzaC on the front and back of the leaves and stem after every 3 d. For the Con group, the plants were treated with clean water. After 30 d of treatment, the 10th to 15th leaves from the bottom of the plant were collected for test. Then the leaves were collected, a small section of 0.6 $cm^2$ in the middle of the leaves was cut along both sides of the main vein and quickly fixed in FAA (90 mL 70% alcohol + five mL formaldehyde + five mL glacial acetic acid) to maintain the original state of the sample. For the observation of anatomical structures, the SOD, POD and CAT activities of another part were immediately measured; the remaining samples were wrapped in tin foil and quick-frozen in liquid nitrogen before being stored in a −80 °C ultra-low temperature refrigerator for determination of remaining physiological indicators and transcriptome sequencing.

### Assessment of changes in morphology

The plant height, stem diameter, root length, biomass, fresh weight of root, and root/shoot ratio of *A. trifoliata* were measured. The aboveground height of plants was measured using a tape measure. The stem diameter was measured using a vernier calipers at a distance of two cm from the ground. Plants were pulled out from the substrate, washed with clean water, and dried at room temperature. Fresh weights of the above- and belowground parts were measured. The root/shoot ratio was calculated as: belowground fresh weight/aboveground fresh weight ×100%.

### Measurement of physiological indexes and sodium content

$Na^+$ content was measured using bromocresol green spectrophotometry (*Xi & Yang, 1998*). Chlorophyll and carotenoid contents were measured using a method reported earlier (*Dai et al., 2009*). The contents of soluble sugar, SOD, $H_2O_2$, and free proline and POD and CAT activities were measured using sulfuric acid–phenol method, nitro-blue tetrazolium photo-oxidation method, titanium sulfate colorimetric method, acidic ninhydrin chromogenic method, guaiacol-$H_2O_2$ chromogenic method, and UV absorption method, respectively (*Gao, 2006*). MDA content was determined using the thiobarbituric acid method (*Turóczy et al., 2011*).

### Assessment of changes in the anatomical structure of root, stem, and leaves of *Akebia trifoliata*

The root, stem, and leaves of plants from the three groups were cut after 30 d of treatment. Root samples were collected from five cm below the rhizome; small leaves in the middle of the ternately compound leaf at the leaf position were collected as the leaf samples, and stem samples was collected from the middle of the stem. All samples were quickly immersed in FAA solution for fixation. Further, the samples were dehydrated with 75%, 85%, 95%, and 100% ethanol for 2 h each and further subjected to treatment with 1/2 anhydrous ethanol + 1/2 xylene (1.5 h), 1/3 anhydrous ethanol + 2/3 xylene (1.5 h), xylene (5 h), and xylene (1 h) to turn the samples transparent. Further, the samples were dipped in wax, sliced,

dewaxed, stained with toluidine blue, decolored, stained with Fast Green FCF, decolorized, and finally examined microscopically using a orthostatic optical microscope (Nikon Eclipse E100, Tokyo, Japan). The images were acquired and data were analyzed using an imaging system (Nikon DS-U3).

## Transcriptome sequencing

Total RNA was extracted from *A. trifoliata* leaves using a total RNA extraction kit (MJZol, Shanghai Major Biomedical Technology Co., Ltd., Shanghai, China). The purity and integrity of RNA were determined using a Nanodrop 2000 spectrophotometer (Thermo Fisher Scientific, Waltham, MA, USA) and agarose gel electrophoresis, respectively. RIN was determined using a biological analyzer (Agilent 5300, Agilent, Santa Clara, CA, USA). After passing the test, the library construction and non-parametric transcriptome sequencing were commissioned to Shanghai Major Biomedical Technology Co., Ltd.

The raw reads obtained from sequencing were subjected to quality control. The reads with adapters, N ratio >10% (indicating that base information could not be determined), all A bases, and low-quality reads (bases with a quality value of Qphred $\leq$ 20 accounting for >50% of the entire reads) were removed so that high-quality clean reads could be obtained for subsequent analysis. Next, Trinity software (https://github.com/trinityrnaseq/trinityrnaseq/wiki) was used to assemble the clean reads to obtain the transcripts. The longest transcript of each gene was taken as the unigene based on the transcript sequence, which was used as the reference sequence for subsequent analyses. The clean reads of each sample were further mapped back to the assembled transcriptome without error using bowtie2 software. 2012). The results of the bowtie2 mapping were further analyzed using RSEM software (http://deweylab.github.io/RSEM/) to quantify the expression level of the genes with the quantitative index FPKM. Subsequently, differential expression analysis was performed using DESeq2 software (http://bioconductor.org/packages/stats/bioc/DESeq2/), and differentially expressed unigenes (DEGs) were screened using the criteria: FDR <0.05 and $|\log_2 FC| \geq 1$. The differences among different groups were compared, and the KEGG pathway enrichment analysis was performed using the DEGs to obtain the biological functions and metabolic pathways that were significantly associated with the DEGs. The metabolic pathway with a corrected $P$ value ($P_{adjust}$) < 0.05 was considered as a significant pathway.

## Data processing

The results were expressed as mean value (from three experimental replicates) $\pm$ SD. Experimental data were organized using Microsoft Excel 2022 software. SPSS 19.0 software (SPSS, Chicago, IL, USA) was used to carry out one-way ANOVA and Duncan's test ($P < 0.05$), whether there is a significant difference three treatment groups in physiological, biochemical, morphological indication were compared. Graphs were plotted using GraphPad Prism 9 and Adobe Illustrator CS6.

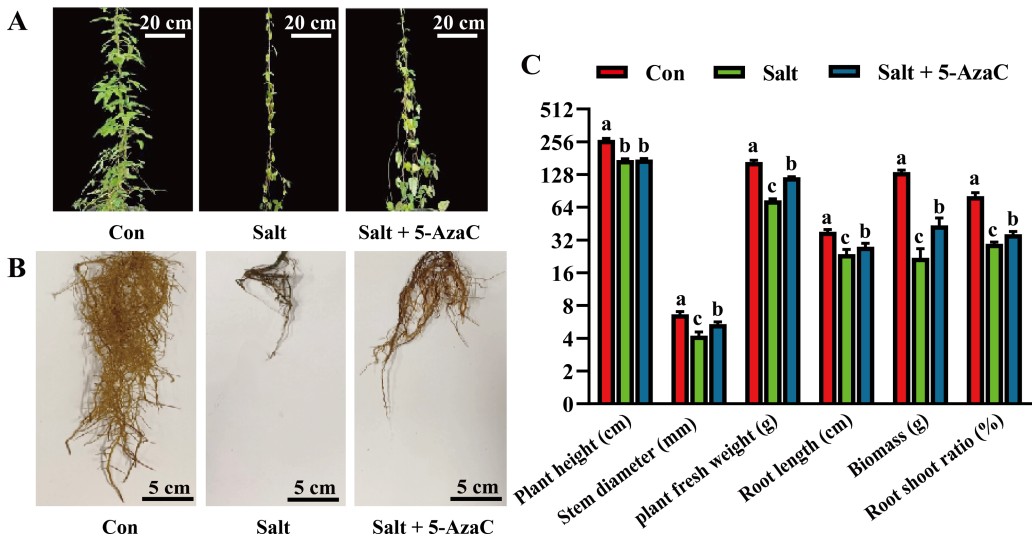

**Figure 1** **The impact of 5-azacytidine (5-AzaC) treatment on the growth of *A. trifoliata* under saline-alkaline stress.** (A) Plant appearance and growth situation of Con, Salt, and Salt+5-AzaC. (B) Root growth situation of Con, Salt, and Salt+5-AzaC. (C) Agronomic traits of Con, Salt, and Salt+5-AzaC. The results are shown as mean ± standard deviation. Different lowercase letters on the bar chart indicate significant differences at the *P* < 0.05 level.

## RESULTS AND ANALYSIS

### Effects of 5-AzaC on the morphology of *Akebia trifoliata* under saline–alkaline stress

To investigate the effects of 5-AzaC on the morphology of *A. trifoliata* under saline–alkaline stress, various morphological indicators were examined. In the Salt group, the leaves were sparse and yellow, indicating a withered state; the main root of the root system was clearly shortened and blackened, and the fibrous root was significantly decayed and decreased compared with the Con group (Fig. 1). The leaf morphology of the Salt+5-AzaC group was normal; the leaves were green; the fibrous roots were fewer and shorter than those of the Con group but more in number and longer than those of the Salt+5-AzaC group. This demonstrated that 5-AzaC treatment could alleviate the dysplasia of the root system caused by saline–alkaline stress. Additionally, the stem diameter, biomass, root length, fresh weight of root, and root/shoot ratio exhibited the trend Con group >Salt+5-AzaC group >Salt group. This demonstrated that saline–alkaline stress treatment affected the stem diameter, biomass, root length, fresh weight of root, and root/shoot ratio. This might be attributed to the fact that saline-alkaline stress leads to reduced lengths of main roots and fibrous roots, which directly affect nutrient uptake and transport, thus affecting the growth and development of the whole plant. However, 5-AzaC could significantly alleviate the symptoms of saline-alkaline stress in *A. trifoliata*.

## Effects of 5-AzaC on the physiological indexes of *Akebia trifoliata* under saline–alkaline stress

Compared with the Con group, $Na^+$ content in the leaves significantly increased and decreased in the Salt and Salt+5-AzaC groups, respectively (Fig. 2), demonstrating that 5-AzaC could reduce the uptake and enrichment of $Na^+$ by plants. The activities of CAT, POD, and SOD were significantly lower in the Salt group than in the Con group. Compared with the Salt group, CAT and POD activities significantly increased in the Salt+5-AzaC group; however, the SOD activity was comparable. Therefore, saline–alkaline stress treatment severely inhibited the activities of CAT, POD, and SOD in the leaves of *A. trifoliata*, whereas 5-AzaC treatment significantly increased CAT and POD activities under saline–alkaline stress. MDA and $H_2O_2$ contents were significantly lower in the Con group than in the Salt group. MDA and $H_2O_2$ contents significantly increased after treatment with 5-AzaC under saline-alkaline stress. Compared with the Con group, soluble sugar content slightly increased and proline content significantly increased in the Salt group, whereas both contents significantly increased in the Salt+5-AzaC group. The contents of chlorophyll a, chlorophyll b, total chlorophyll, and carotenoid significantly reduced in the Salt group. Their contents were significantly higher in the Salt+5-AzaC group than in the Salt group. These results indicated that excessive $Na^+$ entered *A. trifoliata* under saline-alkaline stress, affected its enzyme activity, and interfered with its normal physiological metabolism and life activities, leading to the accumulation of a large amount of toxic substances such as MDA and $H_2O_2$ in the cells. This caused osmotic stress and oxidative damage; reduced the contents of chlorophyll a, chlorophyll b, total chlorophyll, and carotenoid; and affected photosynthesis. However, $Na^+$ restored the equilibrium state after 5-AzaC treatment; this increased the contents of soluble sugar, proline, and photosynthetic pigments and CAT and POD activities compared with saline–alkaline stress conditions, which reduced MDA and $H_2O_2$ contents to a certain extent. Therefore, 5-AzaC treatment could effectively reduce the damage to *A. trifoliata* due to saline–alkaline stress.

## Effects of 5-AzaC on the anatomical structure of *Akebia trifoliata* under saline–alkaline stress

Root cross-section diameter, cortex thickness, phloem thickness, and xylem catheter aperture increased under saline-alkaline stress (Figs. 3A–3C). The root cross-section diameter and xylem catheter aperture significantly increased by 65.92% and 56.80%, respectively, in the Salt+5-AzaC group ($P <0.05$) compared with those in the Salt group. Compared with the Con group, the root cross-section diameter, cortex thickness, phloem thickness, and xylem catheter aperture increased in the Salt group, with the root cross-section diameter and phloem thickness exhibiting significant increase ($P < 0.05$) by 43.50% and 126.72%, respectively, with no significant effect on the xylem and cambium (Fig. 3J). Therefore, spraying of 200 $\mu$mol/L 5-AzaC under saline–alkaline stress could promote lateral root growth and phloem thickening, increase the xylem catheter aperture to enhance the transport efficiency, and weaken the damage due to saline–alkaline stress on the root system of *A. trifoliata*.

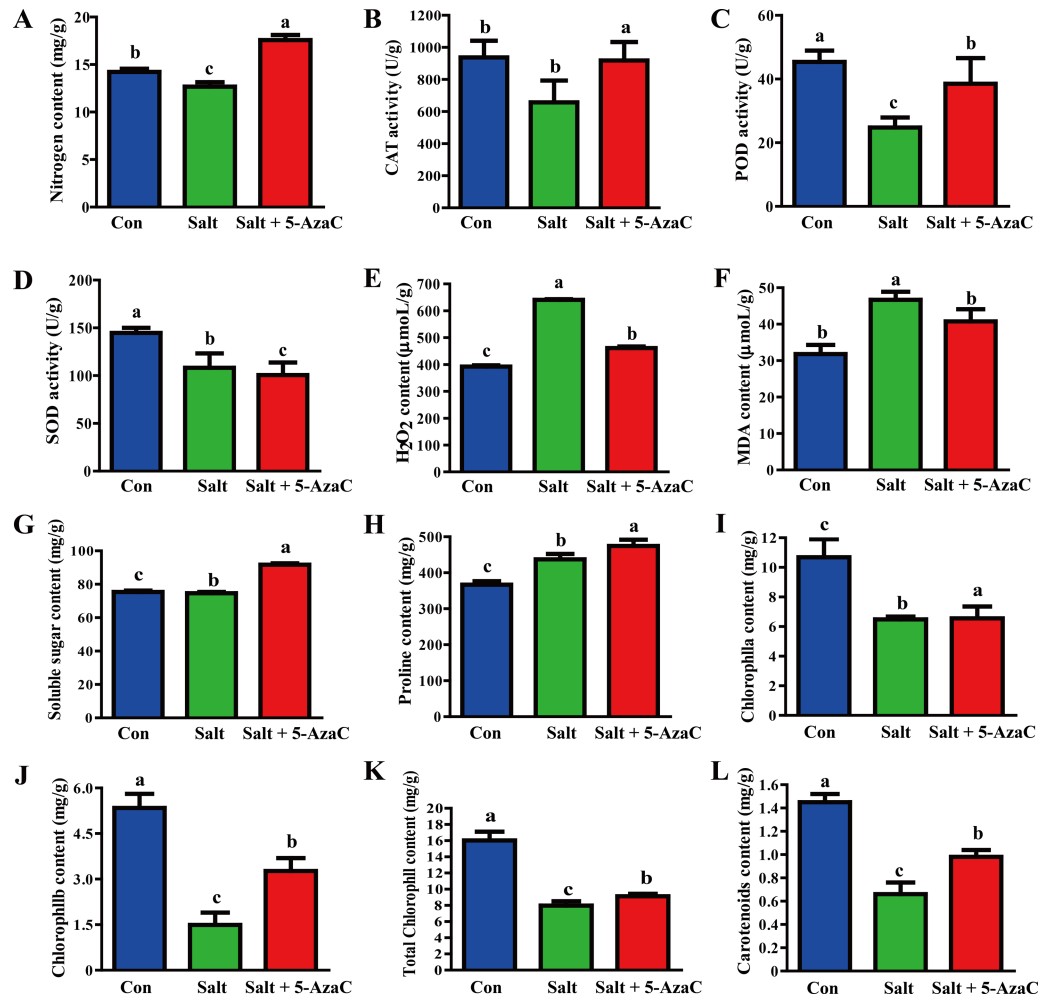

**Figure 2** **Determination of physiological and biochemical indexes in the leaves of *A. trifoliata* treated with 5-AzaC under the saline-alkaline stress.** (A) The contents of Na + in leaves of *A. trifoliata*; (B–D) The activities of CAT, POD, and SOD; (E–H) The contents of $H_2O_2$, MDA, soluble sugar and proline; (I–L) The contents of chlorophyll a, chlorophyll b, total chlorophyll, and carotenoid. The results are shown as mean ± standard deviation. Different lowercase letters on the bar chart indicate significant differences at the *P* < 0.05 level.

Compared with that of the Con group, the stem cambium thickness increased by 50.46% and cortex thickness, phloem thickness, xylem catheter aperture, and medullary radius reduced by 130.68%, 30.73%, 28.05%, and 20.24%, respectively, in the Salt+5-AzaC group. The differences in cortex thickness, phloem thickness, and medullary radius were significant (*P* < 0.05). Compared with the Salt group, the Salt+5-AzaC group exhibited significantly improved cortex thickness, phloem thickness, xylem catheter aperture, and medullary radius (by 49.47%, 49.72%, 55.42%, and 22.12%, respectively) (*P* < 0.05) (Figs. 3D, 3E, 3F, and 3K). Therefore, saline-alkaline stress injured the cortex of *A. trifoliata* and resulted in the thinning of phloem and reduction of xylem catheter aperture and medullary radius, which weakened the transport capacity of the stem and affected the normal physiological

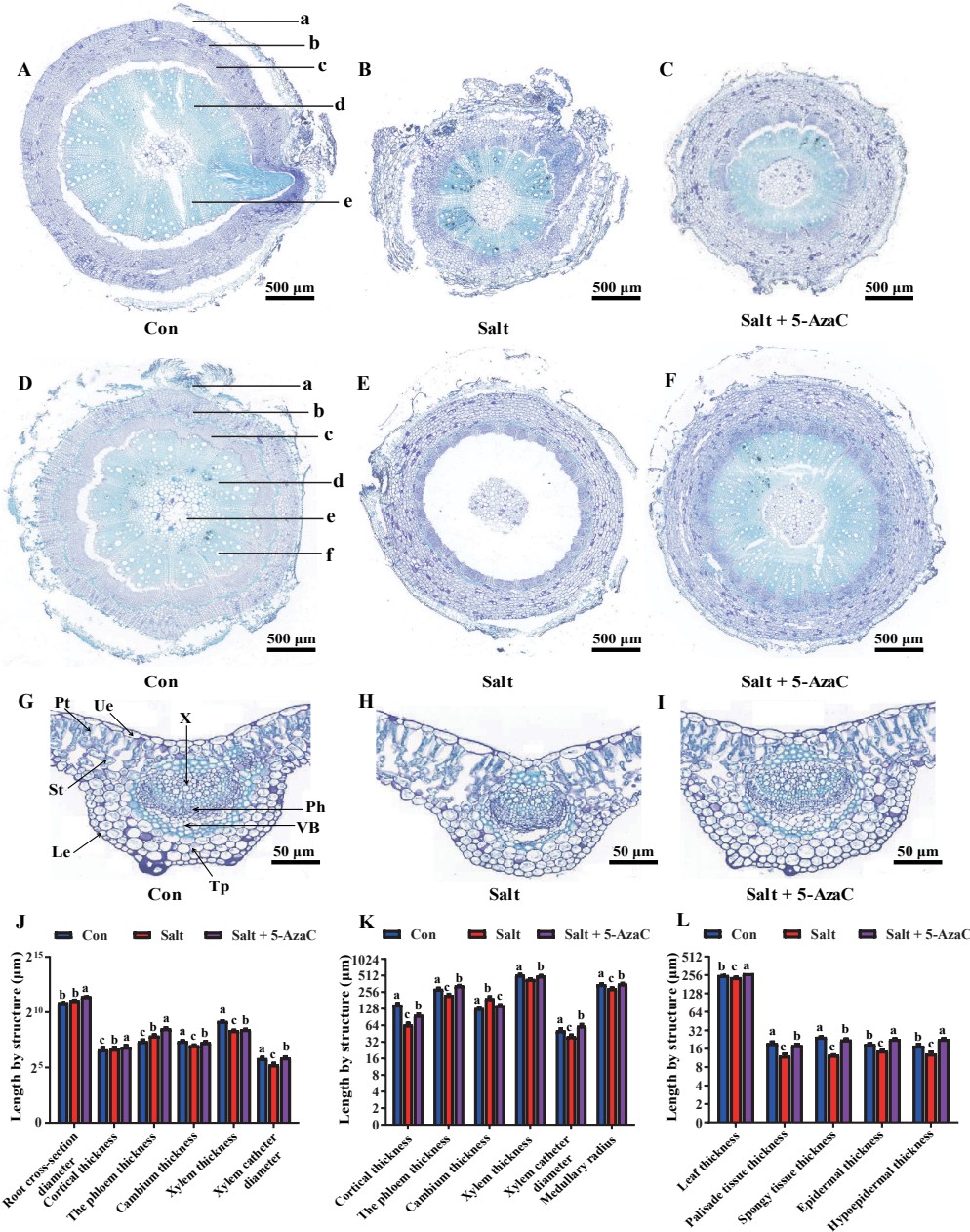

**Figure 3** **Effects of 5-AzaC on the anatomical structure of *A. trifoliata* under saline-alkaline stress.** (A–C) Root structures of *A. trifoliata* in Con (A), Salt (B), and Salt+5-AzaC (C) groups (a: Cortex, b: Phloem, c: Cambium, d: Xylem, e: Xylem vessel). (D–F) Stem structures of *A. trifoliata* in Con (D), Salt (E), and Salt+5-AzaC (F) groups (a: Cortex, b: Phloem, c: Cambium, d: Xylem, e: Xylem vessel, f: Pith). G-I: Leaf structures of *A. trifoliata* in Con (G), Salt (H), and Salt+5-AzaC (I) groups (Ue: Upper epidermis, Le: Lower epidermis, Pt: Palisade parenchyma, St: Spongy tissue, Tp: Thick horn and parenchyma, VB: Vascular bundle, Ph: Phloem, X: Xylem). (J) Changes in root cross-section diameter, cortical thickness, phloem thickness, cambium thickness, xylem thickness, and xylem vessel diameter of *A. trifoliata* in different treatment groups. (K) Changes in cortical thickness, phloem thickness, 

**Figure 3 (…continued)**
cambium thickness, xylem thickness, xylem vessel diameter, and medullary radius of the stem of *A. trifoliata* in different treatment groups. (L) Changes in leaf thickness, palisade tissue thickness, spongy tissue thickness, epidermal thickness, and hypoepidermal thickness of the leaves of *A. trifoliata* in different treatment groups. The results are shown as mean ± standard deviation. Different lowercase letters on the bar chart indicate significant differences at the $P < 0.05$ level.

and metabolic activities. However, spraying of 200 μmol/L 5-AzaC could effectively alleviate this phenomenon and enhance the transport capacity of *A. trifoliata*.

The leaves of *A. trifoliata* are typical bifacial leaves, consisting of upper epidermal cells, lower epidermal cells, palisade cells, and spongy tissue. The thickness of *A. trifoliata* leaves and upper and lower epidermis were reduced under saline-alkaline stress (Fig. 3); Compared with the Con group, the Salt group exhibited decrease by 7.93%, 27.61%, 35.23%, 60.66%, and 97.93%, respectively, whereas the thickness of leaves and upper and lower epidermis exhibited increase by 7.77% and 29.41% compared with the Con group after spraying of 200 μmol/L 5-AzaC. Compared with the Salt group, the Salt+5-AzaC group exhibited significant increase (by 16.31%, 55.56%, 75%, 48.36%, and 78.86%, respectively) in the thickness of leaves, upper epidermis, lower epidermis, palisade cells, and spongy tissue. Additionally, the ratio of palisade cells to spongy tissue was 0.96 in the Salt group, exhibiting increase by 23.08% and 20% compared with the Con and Salt+5-AzaC groups, respectively. This demonstrated that the spongy tissue in leaves of *A. trifoliata* increased in size, and the length of the palisade cells gradually decreased; the tissue layer thinned; the arrangement became looser, and the intercellular space became larger and larger. The cells of the spongy tissue gradually enlarged and changed from compact to lax; advanced air cavities appeared, and interstitial spaces were gradually developed. Overall, spraying of 200 μmol/L 5-AzaC under saline–alkaline stress could alleviate various damages caused by saline-alkaline stress by increasing the thickness of *A. trifoliata* leaves and upper and lower epidermis.

## Effects of 5-AzaC on the transcriptome of *Akebia trifoliata* under saline–alkaline stress

The effect of 5-AzaC on *A. trifoliata* at the gene level was investigated under saline-alkaline stress. Using transcriptome sequencing, 51.84 Gb clean bases and 3.52 G clean reads were obtained from nine samples from three groups of samples with three replicates. The average output of each sample was 6.48 Gb clean bases and 0.44 G clean reads. The GC content of each sample was 42.97%–44.06%, and the percentage of Q20% and Q30% bases exceeded 94% (Table 1). The accuracy of the measured data was high, which met the requirements of quality control and facilitated data analysis in the later stage. Based on the quantitative results, inter-group transcripts analysis was performed to obtain the transcripts between the two groups. The threshold range of $|\log_2 FC| \geq 1.00$ and $P$-adjust $\leq 0.05$ were selected for the analysis of the difference using DESeq2 to identify the DEGs of different groups. The visualization analysis results using a bar chart (Fig. 4B) indicated 8,253, 4,459, and 3,035 transcripts in the Salt *vs* Con, Salt+5-AzaC *vs* Con, and Salt+5-AzaC *vs* Salt groups, respectively. Among them, 3,667, 2,283, and 1,859 transcripts were upregulated and 4,586,

**Table 1  Transcriptome sequencing quality data of *Akebia trifoliata* treated by 5-AzaC under salt and alkali stress.**

| Sample | Clean bases | Clean reads | Q20% | Q30% | GC% |
|---|---|---|---|---|---|
| Con-1 | 6497523443 | 44073089 | 98.17 | 94.86 | 44.10 |
| Con-2 | 6452067836 | 43747844 | 98.17 | 94.78 | 44.06 |
| Con-3 | 6542979049 | 44398334 | 98.17 | 94.87 | 44.15 |
| Salt-1 | 6412546931 | 43599460 | 97.98 | 94.32 | 43.00 |
| Salt-2 | 6390338026 | 43376374 | 98.04 | 94.5 | 42.97 |
| Salt-3 | 6128863841 | 41613048 | 98.18 | 94.87 | 43.23 |
| Salt+5-AzaC-1 | 6681928214 | 45332346 | 98.06 | 94.56 | 43.51 |
| Salt+5-AzaC-2 | 6610329300 | 44717082 | 98.09 | 94.55 | 43.95 |
| Salt+5-AzaC-3 | 6625871183 | 45157322 | 97.99 | 94.42 | 43.36 |

**Notes.**

Note: (1) Sample: Sample name; (2) Clean bases: the total amount of sequencing data after quality control; (3) Q20 (%), Q30 (%): Evaluate the sequencing data after quality control. Q20 and Q30 refer to the percentage of bases sequenced at more than 99% and 99.9% of the total bases respectively. Generally, Q20 is above 85%, Q30 is above 80%; (4) GC content (%): The sum of G and C bases corresponding to the quality control data as a percentage of the total bases.

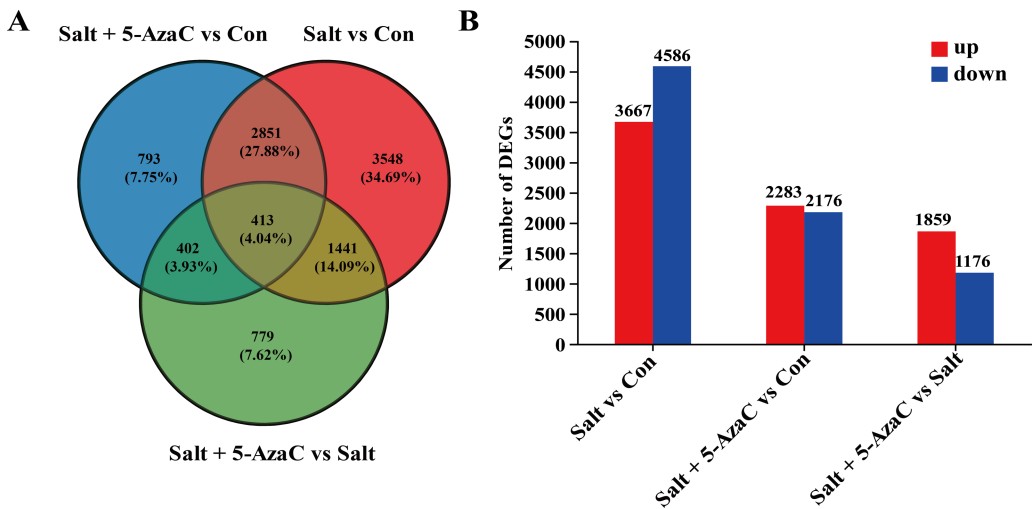

**Figure 4  Effect of 5-AzaC on *A. trifoliata* differentially expressed unigenes (DEGs) under saline-alkali stress.** (A) Venn analysis of differential genes among Salt+5-AzaC *vs* Con, Salt *vs* Con, and Salt+5-AzaC *vs* Salt. (B) Histogram of differential gene expression analysis between Salt *vs* Con, Salt+5-AzaC *vs* Con, and Salt+5-AzaC *vs* Salt. The *x*-axis indicates comparison groups; the *y*-axis indicates the number of genes. Red and blue bars represent upregulated and downregulated genes, respectively.

2,176, and 1,176 transcripts were downregulated. A total of 3,264, 815, and 1,854 transcripts were observed between the Salt+5-AzaC *vs* Con and Salt *vs* Con, Salt+5-AzaC *vs* Con and Salt+5-AzaC *vs* Salt, and Salt *vs* Con and Salt+5-AzaC *vs* Salt groups, respectively. In total, 413 transcripts were observed in the three groups. Overall, 793, 3,548, and 779 unique transcripts were observed in the Salt+5-AzaC *vs* Con, Salt *vs* Con, and Salt+5-AzaC *vs* Salt groups, respectively (Fig. 4A). The shared genes may have occurred due to the common environment, whereas the unique ones may be in response to different treatments.

### KEGG pathway enrichment analysis of the DEGs involved in the effects of 5-AzaC on *Akebia trifoliata* under saline–alkaline stress

In Salt *vs* Con, 935 transcripts were enriched in the KEGG pathway (Fig. 5). Among them, plant hormone signal transduction (map04075), oxidative phosphorylation (map00190), amino sugar and nucleotide sugar metabolism (map00520), phenylpropanoid biosynthesis (map00940), photosynthesis (map00195), sesquiterpenoid and triterpenoid biosynthesis (map00909), fatty acid biosynthesis (map00061), and fatty acid elongation (map00062) had the highest number of enriched transcripts ($n = 116$, 84, 58, 54, 41, 23, 21, and 18, respectively). In Salt+5-AzaC *vs* Con, 1462 transcripts were enriched in the KEGG pathway. The pathways with the highest number of transcripts (60, 45, and 37) were plant hormone signal transduction (map04075), amino sugar and nucleotide sugar metabolism (map00520), and phenylpropanoid biosynthesis (map00940), respectively. In Salt+5-AzaC *vs* Salt, 935 transcripts were enriched in the KEGG pathway with the highest number of genes (28, 12, 23, and 6) enriching in photosynthesis (map00195), photosynthesis - antenna proteins (map00196), glutathione metabolism (map00480), and flavone and flavonol biosynthesis (map00944), respectively.

The transcripts between Salt+5-AzaC *vs* Salt and Salt+5-AzaC *vs* Con were mainly enriched in metabolic pathways. The number of transcripts was higher in Salt+5-AzaC *vs* Con than in Salt+5-AzaC *vs* Salt. This suggested that 5-AzaC could stimulate more metabolic response mechanisms under saline-alkaline stress in *A. trifoliata*, enhance the metabolic capacity of plants, and thus alleviate the damage caused by saline-alkaline stress to *A. trifoliata*, promoting the development of *A. trifoliata*.

### Key genes in *Akebia trifoliata* after 5-AzaC treatment under saline–alkaline stress
#### Genes involved in plant hormone signal transduction
KEGG pathway enrichment analysis indicated that in Salt *vs* Con and Salt+5-AzaC *vs* Con, the transcripts were highly enriched in plant hormone signal transduction (Figs. 5A and 5B). Notably, most transcripts in plant hormone signal transduction belonged to auxin synthetic pathway. Compared with the Con group, 65 transcripts in the Salt group were involved in auxin synthesis. Among them, 2, 2, 12, 44, and one transcripts were downregulated, which were responsible for encoding auxin transporter-like protein 3 (AUX1), auxin-responsive promoter (GH3), hypothetical protein (IAA), hypothetical protein (SUAR), and transport inhibitor response 1 protein (TIR1), respectively (Table S1, Fig. 6). This suggested that saline–alkaline stress can significantly inhibit auxin synthesis in *A. trifoliata*. Compared with the Salt group, the Salt+5-AzaC group contained 10 DEGs, of which 9 were upregulated. The 6 DEGs that are responsible for encoding SAUR were downregulated in the Salt group compared with the Con group. These six transcripts were upregulated in Salt+5-AzaC *vs* Salt. This demonstrated that 5-AzaC may regulate the auxin pathway in *A. trifoliata* by regulating the expression of *SAUR*, thus promoting growth of *A. trifoliata*, and improving its saline–alkaline resistance, which was consistent with the morphological and leaf anatomical observations of *A. trifoliata* after 5-AzaC treatment.

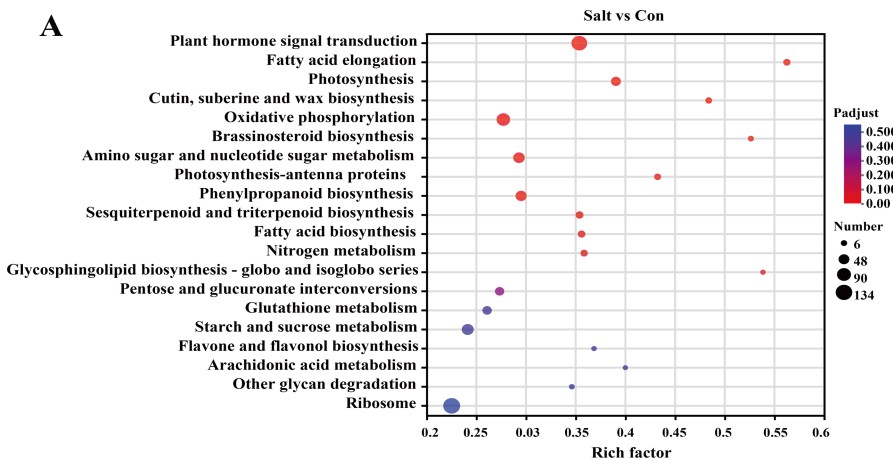

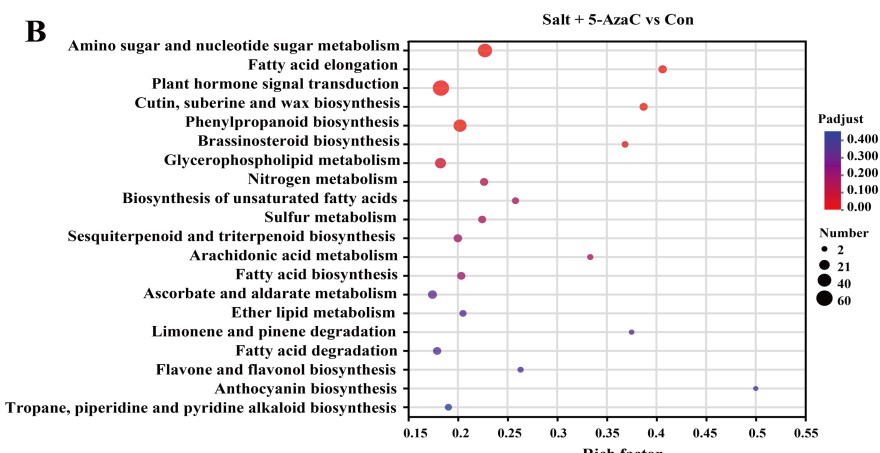

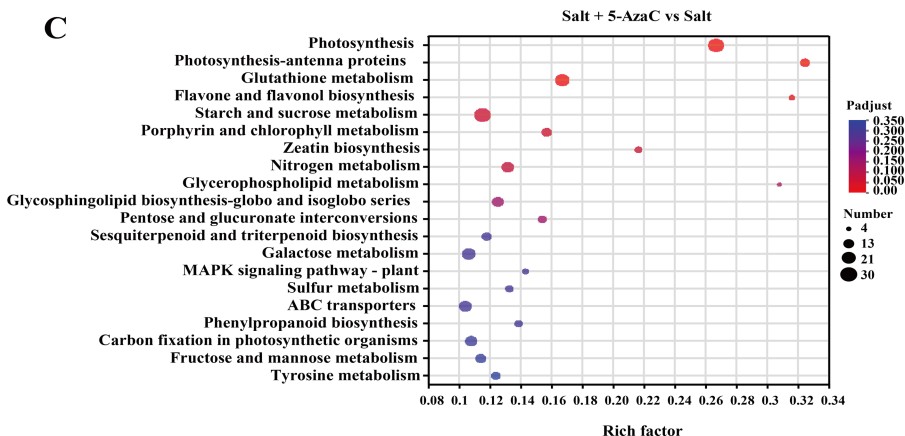

**Figure 5** **KEGG enrichment analysis of the effects of 5-AzaC on DEGs of *A. trifoliata* under saline-alkaline stress.** (A–C) The top 20 pathways enriched by KEGG in the groups of Salt *vs* Con, Salt+5-AzaC *vs* Con, and Salt+5-AzaC *vs* Salt. A significant enrichment is indicated when the *P* adjust < 0.05.

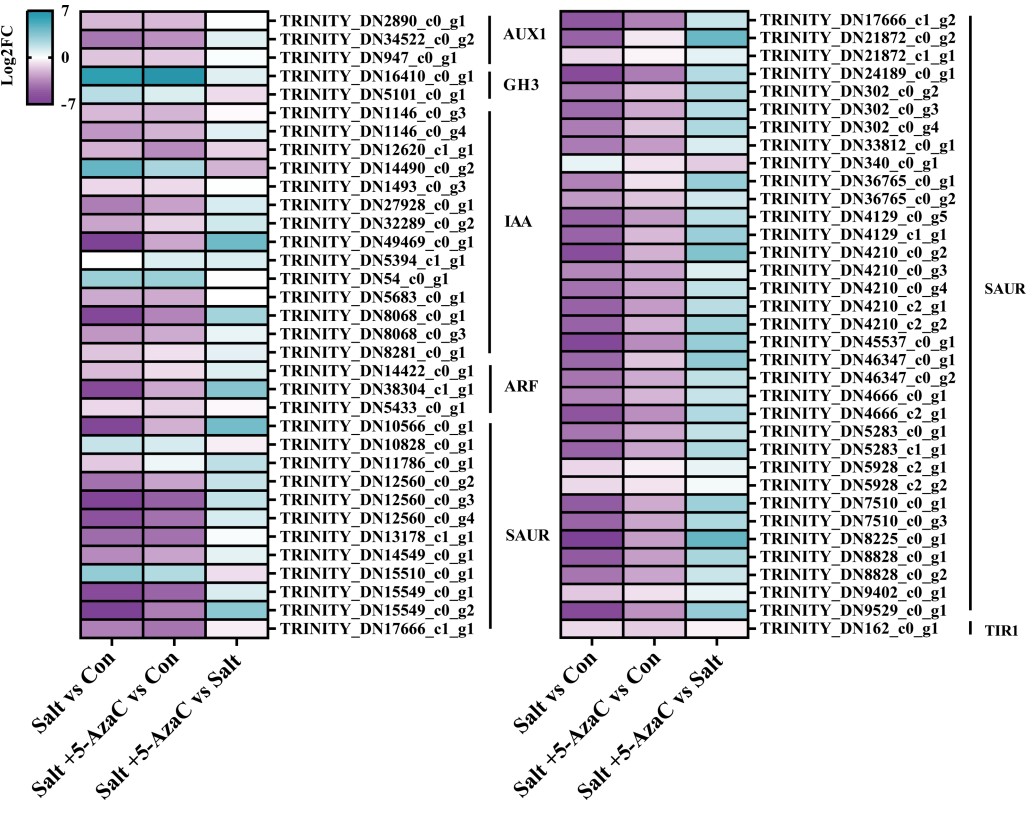

**Figure 6 DEGs involved in plant hormone signal transduction.** AUX1, auxin influx carrier; GH3, auxin responsive GH3 gene family; IAA, auxin-responsive protein IAA; ARF, auxin response factor; SAUR, SAUR family protein; TIR1, transport inhibitor response 1.

### Transcripts involved in phenylpropanoid biosynthesis

Phenylpropanoid biosynthesis was also significantly enriched in Salt *vs* Con and Salt+5-AzaC *vs* Con (Figs. 5A and 5B). Several unigenes were involved in phenylpropanoid biosynthesis, including 4-coumarate-CoA ligase (4CL), beta-glucosidase (E3.2.1.21), cinnamyl-alcohol dehydrogenase (CAD), caffeic acid 3-O-methyltransferase (COMT), caffeoylshikimate esterase (CSE), peroxidase (E1.11.1.7), caffeoyl-CoA O-methyltransferase (E2.1.1.104), shikimate O-hydroxycinnamoyltransferase (HCT), glycoside hydrolase (E3.2.1.21), coniferyl-aldehyde dehydrogenase (REF1), scopoletin glucosyltransferase (TOGT1), and coniferyl-alcohol glucosyltransferase (UGT72E) (Table S2, Fig. 7). Compared with the Con group, the Salt group had 54 transcripts. Among them, 14, 12, 2, 1, 2, 1, 1, 1, 1, and 1 transcripts that are responsible for encoding E1.11.1.7, E3.2.1.21, HCT, E2.1.1.104, 4CL, REF1, CSE, COMT, CAD, and TOGT1, respectively, were downregulated. Meanwhile, 4, 2, 5, 1, 1, 1, 1, and 1 DEGs that are responsible for encoding E3.2.1.21, TOGT1, E1.11.1.7, HCT, CSE, CAD, 4CL, and UGT72E, respectively, were upregulated (Table S2, Fig. 7). This suggested that most unigenes involved in phenylpropanoid biosynthesis were downregulated under saline–alkaline stress. Compared with the Con group, the Salt+5-AzaC group had 10 new transcripts. Among them, eight

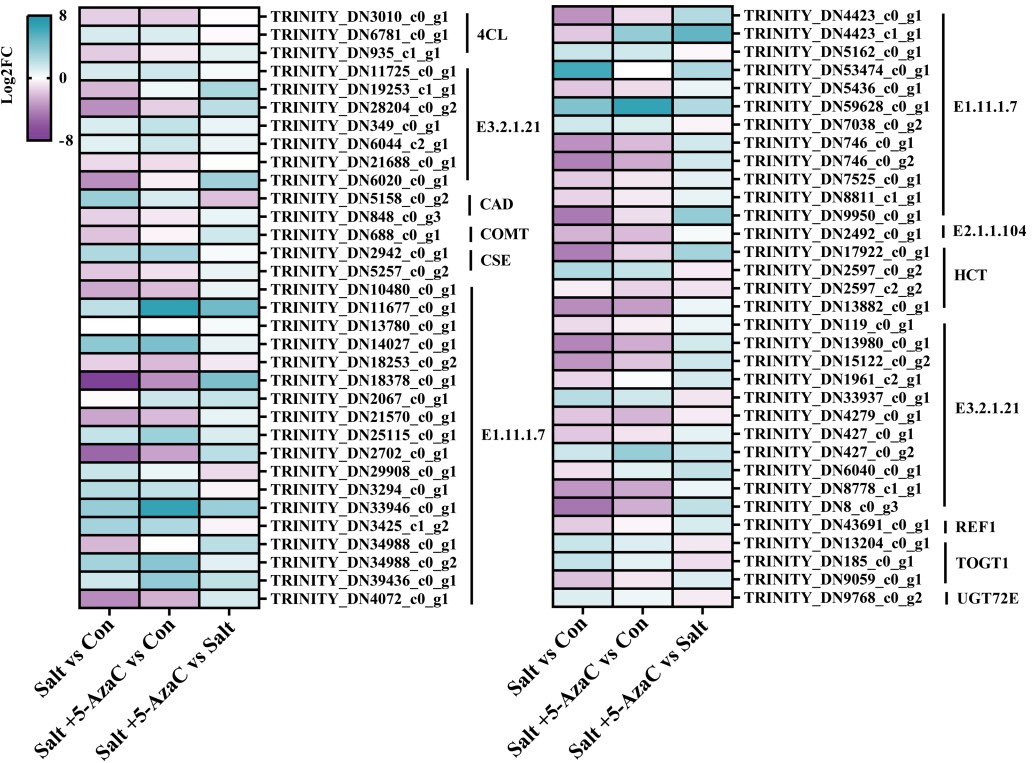

**Figure 7 DEGs involved in phenylpropanoid biosynthesis.** E1.11.1.7: peroxidase; E2.1.1.104: caffeoyl-CoA-O-methyltransferase; HCT: shikimate O-hydroxycinnamoyltransferase; E3.2.1.21: beta-glucosidase; REF1: coniferyl-aldehyde dehydrogenase; TOGT1: scopoletin glucosyltransferase; UGT72E: coniferyl-alcohol glucosyltransferase; 4CL: 4-coumarate-CoA ligase; CAD: cinnamyl-alcohol dehydrogenase; COMT: caffeic acid 3-O-methyltransferase; CSE: caffeoylshikimate esterase.

and one transcripts that are responsible for encoding E1.11.1.7 and E3.2.1.21, respectively, were upregulated (Table S2, Fig. 7). This suggested that 5-AzaC may promote POD activity under saline–alkaline stress by specifically promoting the expression of E1.11.1.7, thereby enhancing the antioxidant capacity of *A. trifoliata*. This was consistent with the significant increase in POD activity after 5-AzaC treatment as described in subsection 'Effects of 5-AzaC on the physiological indexes of *Akebia trifoliata* under saline–alkaline stress'. Additionally, compared with the Salt group, the Salt+5-AzaC group had 19 transcripts. Among them, 18 were upregulated, and 5, 5, 1, 1, and 1 transcripts that are responsible for encoding E3.2.1.21, E1.11.1.7, REF1, COMT, and TOGT1, respectively, were downregulated in Salt *vs* Con (Table S2, Fig. 7). This indicated that the enhanced expression of these unigenes may be related to the potential mechanism of 5-AzaC in enhancing saline–alkaline resistance of *A. trifoliata*.

### DEGs participating in photosynthesis

Photosynthesis was the most significantly enriched KEGG pathway in Salt+5-AzaC *vs* Salt (Fig. 5C). Many unigenes involved in photosynthesis were affected by 5-AzaC. Compared with the Con group, the Salt group had 41 transcripts, of which 31 were down-regulated

(Table S3, Fig. 8). Of these 31 transcripts, 1, 3, 1, 2, 2, 1, and 1 transcripts encoding hypothetical protein (psbA), oxygen-evolving enhancer protein 1 (psbQ), photosystem II family protein (psbS), photosystem II reaction center W protein (psbW), photosystem II PsbY (psbY), hypothetical protein (psb27), and hypothetical protein (psb28) related to photosystem II; 1, 1, 2, 1, 1, and 1 DEGs encoding hypothetical protein (psaD), hypothetical protein (psaE), photosystem I PsaG/PsaK protein (psaG), hypothetical protein (psaH), photosystem I reaction center subunit psaK (psaK), and photosystem I PsaL (psaL) related to photosystem I; 1 and 5 transcripts encoding plastocyanin (petE) and ferredoxin (petF) related to photosynthetic electron transport; and 1, 1, and 1 transcripts encoding hypothetical protein (atpF) , ATPase ( atpG), and ATP synthase delta chain (atpH) related to F-ATPase, respectively, were downregulated in the Salt group (Table S3, Fig. 8). However, these transcripts were upregulated in Salt *vs* Salt+5-AzaC (Table S3, Fig. 8). Therefore, photosynthesis of *A. trifoliata* was significantly inhibited by saline-alkaline stress, and this inhibition was ameliorated by 5-AzaC treatment.

## DISCUSSION

Under saline-alkaline stress, the excessive accumulation of $Na^+$ can easily lead to the damage of the cell membrane system, thus inducing oxidative stress (*Zhu, 2003*). The results of *Zhong, Xu & Wang (2010)* indicated that the growth of wheat seedlings was inhibited under salt stress the root $Na^+$ content was increased, and the addition of exogenous 5-AzaC significantly reduced the growth inhibition of wheat, and at the same time, the root $Na^+$ content was also significantly reduced. In this study, compared with saline-alkali treatment alone, 5-AzaC treatment effectively reduced the $Na^+$ uptake by *A. trifoliata* and reduced the cellular damage caused by high $Na^+$ concentration, which helped to improve its resistance to saline-alkaline stress. In addition, osmoregulation plays an important role in plant stress resistance, mainly regulating plant stress in two aspects, on the one hand through osmotic pressure regulation function to improve plant water absorption capacity, on the other hand through osmotic protection to maintain the integrity of cell structure and function (*Hans et al., 2006*). Soluble sugar and proline are important osmoregulatory substances in cells. The accumulation of soluble sugar and proline can regulate cell osmotic pressure and enhance plant's ability to adapt to stress (*Maach et al., 2020*). The results of this study showed that 5-AzaC treatment could increase the contents of proline and soluble sugar in leaves of *A. trifoliata* seedlings under salt-alkali stress. This is consistent with the previous reports (*Li et al., 2020*), indicating that under saline-alkali stress, 5-AzaC can promote the synthesis of osmotic adjustment substances, thereby maintaining osmotic pressure balance, and further maintaining the integrity of cell structure and function.

Chlorophyll is an important pigment for plant photosynthesis, its content is an important basis for measuring plant growth and development and evaluating plant physiological status, and is an important substance involved in the process of light reaction (*Gharibiyan et al., 2023*). In this study, physiological indicators showed that salt treatment significantly decreased the chlorophyll content, the integrity of the structure of the blade fence is damaged, it may be related to the disintegration and inactivation of plant chloroplasts

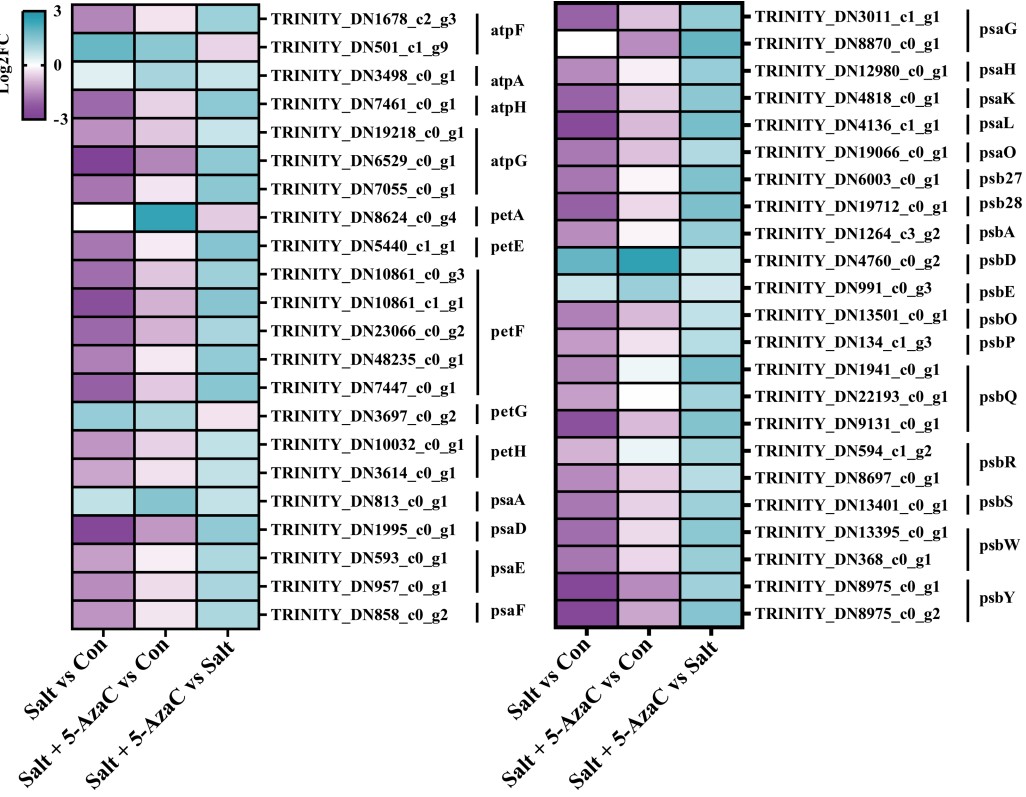

**Figure 8 DEGs participating in photosynthesis.** atpF: F-type H + -transporting ATPase subunit b; atpA: F-type H + /Na + -transporting ATPase subunit alpha ; atpH: F-type H + -transporting ATPase subunit delta; atpG: F-type H + -transporting ATPase subunit gamma; petA: apocytochrome f; petE: plastocyanin; petF: ferredoxin; petG: cytochrome b6-f complex subunit 5; petH: ferredoxin-NADP+ reductase; psaA: photosystem I P700 chlorophyll a apoprotein A1; psaD: photosystem I subunit II; psaE: photosystem I subunit IV; psaF: photosystem I subunit III; psaG: photosystem I subunit V; psaH: photosystem I subunit VI; psaK: photosystem I subunit X; psaL: photosystem I subunit XI; psaO: photosystem I subunit; psaO; psb27: photosystem II Psb27 protein; psb28: photosystem II 13kDa protein; psbA: photosystem II P680 reaction center D1 protein; psbD: photosystem II P680 reaction center D2 protein; psbE: photosystem II cytochrome b559 subunit alpha; psbO: photosystem II oxygen-evolving enhancer protein 1; psbP: photosystem II oxygen-evolving enhancer protein 2; psbQ: photosystem II oxygen-evolving enhancer protein 3; psbR: photosystem II 10kDa protein; psbS: photosystem II 22kDa protein; psbW: photosystem II PsbW protein; psbY: photosystem II PsbY protein.

under saline-alkali stress, while compared with the salt treatment, salt + 5-AzaC significantly increased the total chlorophyll content of leaves and the palisade tissue structure remained intact and the integrity of palisade tissue was closely related to photosynthetic capacity. Meanwhile, also confirmed that exogenous 5-AzaC treatment alleviated the salt stress on kenaf, significantly increasing the chlorophyll content of seedlings while reducing the production of ROS (*Li et al., 2024*). Thus, it is speculated that saline-alkali stress may cause damage to the plant roots, stem vessels, and leaf tissues, thereby hindering chlorophyll synthesis, however exogenous 5-AzaC can act as an inhibitor of photosynthetic pigment degradation, exerting the effect of protecting the structure and function of the photosynthetic system.

Under saline-alkaline stress, $H_2O_2$ and MDA can easily disrupt the metabolic balance of intracellular ROS, and the large amount of accumulated ROS can cause severe peroxidative damage to the plasma membrane (*Jiang et al., 2019*). SOD, POD, and CAT are the key indicators for measuring plant stress resistance and the increased activity of them can reduce the damage caused by adversity stress (*Gill & Tuteja, 2010*). *Yao et al. (2023)* found that 5-AzaC treatment significantly enhanced the POD and CAT activities of tomato seedlings under salt stress. Similar to that results , in this study, compared with the Con group, $H_2O_2$ and MDA contents in the leaves significantly increased and SOD, activities of POD and CAT significantly decreased under saline-alkaline stress in the Salt group. This demonstrated that saline-alkaline stress caused serious oxidative damage to the cell membrane of *A. trifoliata*. The total chlorophyll content and POD and CAT activities were significantly higher in the Salt + 5-AzaC group than in the Salt group. It was evident that 5-AzaC could improve the saline-alkali resistance of *A. trifoliata* seedlings by increasing the activities of different antioxidant enzymes, which was similar to the results of previous studies (*Li et al., 2024*).

The result showed that many genes involved in plant hormone signal transduction, phenylpropanoid biosynthesis and photosynthesis pathways are regulated by 5-AzaC. In plant hormone signal transduction pathway, most genes participate in auxin synthesis, among of these genes, AUX/IAA and SAUR are the two most important early auxin-responsive gene families (*He, 2019*). The expression pattern of AUX/IAA gene family is relatively complex. For example, some AUX/IAA-encoding genes are involved in the differentiation of *Dimocarpus longan* flower bud (*Jue et al., 2019*), and they regulate drought resistance in *Arabidopsis thaliana* (*Salehin et al., 2019*). As the largest gene family among the early auxin-responsive genes in plants (*Shin et al., 2019*), SAUR plays a key role in the resistance of plants to biotic and abiotic stresses. For example, overexpression of SAUR-encoding genes can lead to increased survival rates of *Triticum aestivum* under drought and salt stresses (*Guo et al., 2018*). In this study, most transcripts related to the auxin pathway belonged to the AUX/IAA and SAUR families, demonstrating that the auxin pathway plays a key role in the resistance of plants to saline-alkaline stress. The result of transcripts analysis that most of the auxin pathway related DEGs belong to the IAA and SAUR families, and most of these DEGs were down-regulated in the Salt *vs* Con comparison, but 5-AzaC treatment increased their expression. The researches show that 5-AzaC can regulate gene expression in plant hormone signal transmission pathway, especially auxin regulation pathway in embryogenic callus of longan (*Chen et al., 2020a*). Therefore, we speculate that the auxin regulatory pathway in the plant hormone signal transmission pathway may be a key regulatory factor in the 5-AzaC-mediated increase in saline-alkali resistance of *Akebia trifoliata*. The results corresponded to the plant growth after spraying 5-AzaC under saline-alkali stress.

The roles of phenylpropane biosynthetic pathway in the resistance of plants to abiotic stress have been explored (*Zhou et al., 2016b*; *Zou et al., 2019*). In this study, saline-alkaline stress inhibited the expression of most genes in the phenylpropanoid biosynthesis pathway, but 5-AzaC treatment increased their expression. It is worth noting that the Salt + 5-AzaC group had 19 transcripts (with 18 upregulated transcripts) compared with the Salt group.

Five transcripts encoding E1.11.1.7 were down-regulated in Salt *vs* Con but upregulated in Salt + 5-AzaC *vs* Salt. Hence, 5-AzaC may regulate the saline-alkaline resistance of *A. trifoliata* under saline-alkaline stress by regulating the expression of E1.11.1.7. This is consistent with significantly increased POD activity in *A. trifoliata* leaves. Additionally, E1.11.1.7 expression can significantly affect lignin biosynthesis (*Qiu et al., 2021*), however, the accumulation of lignin in plants can increase the strength of plant cells and enhance the tolerance of plant stress (*Dong et al., 2023*), which is of great significance as lignin improves plant stress resistance and tolerance (*Sharma et al., 2020*). In this study, compared with saline-alkali stress alone, the thickness of upper epidermis, lower epidermis and palisade tissue of *Akebia trifoliata* seedlings treated with 5-AzaC was significantly increase and resulting in increased blade thickness. At the same time, xylem, xylem duct pore diameter and phloem thickness of roots and stems were also significant increase. The results showed that leaf thickness was conducive to water transpiration, xylem vessel pore size directly affected the transport of radial water flow and water use efficiency, phloem acted as a barrier to isolate harmful ions, and the enhancement of palisade tissue, upper epidermis, lower epidermis, xylem and phloem was related to the accumulation of lignin (*Dong & Zhang, 2001*). Moreover, lignin accumulation facilitates the activation of plant defense systems in response to selective pressures from the environment (*Yadav et al., 2020*). Therefore, we speculated that 5-AzaC treatment may improve the mechanical strength of the seedlings by regulating lignin biosynthesis, thereby reducing the damage of harmful ions to the plants under salt and alkali stress.

The effects of saline-alkaline stress on photosynthesis of plants have been thoroughly investigated (*Ren et al., 2023*; *Ling et al., 2022*). The study found that 5-azaC treatment increased the expression of genes in the photosynthesis metabolic pathway of bitter melon, thereby affecting the rate at which chloroplasts in the pulp tissue transform into chromoplasts, thereby affecting its maturation process (*Guo et al., 2023*). This shows that 5-AzaC also has potential in regulating plant photosynthesis effect. In this study, various unigenes involved in the KEGG pathway of photosynthesis, including photosystem II (PSII), photosystem I (PSI), photosynthetic electron transport, and F-ATPase, were affected by 5-AzaC. Genes involved in the photosynthesis process have mostly been down-regulated in the Salt *vs* Con comparison, but 5-AzaC treatment improves their expression. Under saline-alkaline stress, $Na^+$-induced oxidative stress disrupts the reaction center of PSII in plants, resulting in degraded activity of PSII (*Zhang et al., 2002*). Meanwhile, ROS accumulated in the chloroplast mainly inhibits the repair process of PSII by inhibiting protein synthesis in the PSII and mainly targets PsbA-encoded D1 protein (*Chen et al., 2020b*). In this study, PsbA-encoding transcripts were upregulated and chlorophyll content significantly increased and the tissue structure of leaf palisade remained intact after 5-AzaC treatment compared with saline-alkaline stress treatment alone. This suggested that 5-AzaC significantly positively affected stability maintenance and accumulation of photosynthetic pigments in PSII of *A. trifoliata* under saline-alkaline stress. The iron-sulfur cluster in PSI receptor is the first site attacked by ROS. Abundant ROS scavengers (SOD and APX) are distributed in the vicinity to prevent ROS from spreading to the substrate and causing toxicity (*Zhang, Yang & Gao, 2013*). The results indicated that SOD and

POD activities were significantly reduced by salt-alkali treatment alone, and transcripts encoding psaD, psaE, psaG, psaH, psaK, and psaL were significantly down-regulated in the Salt *vs* Con comparison, witch indicating that salt-alkali stress may inhibit the activity of ROS scavenging enzymes. However, after 5-AzaC treatment, POD and CAT activities were significantly increased, and transcripts encoding psaD, psaE, psaG, psaH, psaK, and psaL were significantly upregulated. As electron carriers on the electron transport chain, plastocyanin and ferredoxin play a key role in photosynthetic electron transport and energy conversion (*Magistretti, 1987*). In this study, PetE-encoding transcripts in the plastocyanin protein and PetF-encoding transcripts in the ferredoxin protein were down-regulated under saline-alkaline stress compared with the Con group. After 5-AzaC treatment, these transcripts were upregulated, indicating that 5-AzaC-induced expression of petF may play a key role in reducing toxicity related to saline-alkaline stress in plants. F-ATPase located in the thylakoid membrane of chloroplast can catalyze the conversion of ADP to ATP, thereby providing energy for carbon fixation by plants (*Kühlbrandt, 2019*). In this study, transcripts such as atpF, atpG, and atpH in the ATP synthetic pathway were down-regulated under saline-alkaline stress but upregulated after 5-AzaC treatment. This indicated that 5-AzaC may promote ATP synthesis and provide energy for carbon assimilation. Overall, 5-AzaC has a potential role in protecting *A. trifoliata* from saline-alkaline stress. However, further studies are needed to explore more relevant signal transduction pathways involved in the action of 5-AzaC in enhancing the saline-alkaline resistance of plants. In this study, PetE-encoding transcripts in the plastocyanin protein and PetF-encoding transcripts in the ferredoxin protein were down-regulated under saline-alkaline stress compared with the Con group. After 5-AzaC treatment, these transcripts were upregulated, indicating that 5-AzaC-induced expression of *petF* may play a key role in reducing toxicity related to saline-alkaline stress in plants. F-ATPase located in the thylakoid membrane of chloroplast can catalyze the conversion of ADP to ATP, thereby providing energy for carbon fixation by plants (*Kühlbrandt, 2019*). In this study, transcripts such as atpF, atpG, and atpH in the ATP synthetic pathway were down-regulated under saline-alkaline stress but upregulated after 5-AzaC treatment. This indicated that 5-AzaC may promote ATP synthesis and provide energy for carbon assimilation. Overall, 5-AzaC has a potential role in protecting *A. trifoliata* from saline-alkaline stress. However, further studies are needed to explore more relevant signal transduction pathways involved in the action of 5-AzaC in enhancing the saline-alkaline resistance of plants.

## CONCLUSION

Saline-alkaline stress severely inhibited the growth and development of *A. trifoliata*; however, exogenous 5-AzaC application could effectively alleviate the negative impacts of saline–alkaline stress on *A. trifoliata* in terms of stem diameter, root length, fresh weight of root, root/shoot ratio, and biomass. Specifically, 5-AzaC could increase the activity of antioxidant enzymes and content of osmoregulatory substances in *A. trifoliata* under saline–alkaline stress. $H_2O_2$ and MDA contents were reduced after 5-AzaC treatment, thus reducing osmotic stress and oxidative damage of *A. trifoliata* under saline–alkaline stress.

Additionally, 5-AzaC could promote chlorophyll synthesis in *A. trifoliata* and protect photosynthetic systems from damage under saline–alkaline stress. Moreover, it could effectively reduce $Na^+$ uptake and regulate ionic homeostasis. Meanwhile, the increased expression of genes from auxin pathway in plant hormone signal transduction; lignin synthetic pathway in phenylpropanoid biosynthesis; and PSII, PSI, photosynthetic electron transport, and F-ATPase pathway in photosynthesis may be closely related to 5-AzaC-induced saline–alkaline resistance of *A. trifoliata*. This study provided references for using 5-AzaC to enhance the saline-alkaline resistance of *A. trifoliata* in practical applications.

## ACKNOWLEDGEMENTS

The authors are grateful to the editors and the reviewers for their valuable comments and help. This work received valuable laboratory site support from Kunming University.

### Funding

This work was supported by the National Natural Science Foundation of China (32060645), the Joint Special Project (Key Project) of Local Universities in Yunnan Province (202101BA070000-036), Yunnan Students' innovation and entrepreneurship training program (S202311393052; S202411393039; S202411393055) and the Joint Special Project (Surface Project) of Yunnan Province Local Undergraduate University (202101BA070001-172). The funders had no role in study design, data collection and analysis, decision to publish, or preparation of the manuscript.

### Grant Disclosures

The following grant information was disclosed by the authors:
National Natural Science Foundation of China: 32060645.
Joint Special Project (Key Project) of Local Universities in Yunnan Province: 202101BA070000-036.
Yunnan Students' innovation and entrepreneurship training program: S202311393052, S202411393039, S202411393055.
Joint Special Project (Surface Project) of Yunnan Province Local Undergraduate University: 202101BA070001-172.

### Competing Interests

The authors declare there are no competing interests.

### Author Contributions

- Xiao Xu Bi conceived and designed the experiments, performed the experiments, analyzed the data, prepared figures and/or tables, authored or reviewed drafts of the article, and approved the final draft.
- Kai Wang conceived and designed the experiments, performed the experiments, analyzed the data, prepared figures and/or tables, authored or reviewed drafts of the article, and approved the final draft.

- Xiaoqin Li conceived and designed the experiments, performed the experiments, analyzed the data, prepared figures and/or tables, authored or reviewed drafts of the article, and approved the final draft.
- Jiao Chen conceived and designed the experiments, performed the experiments, analyzed the data, prepared figures and/or tables, authored or reviewed drafts of the article, and approved the final draft.
- Jin Yang conceived and designed the experiments, performed the experiments, analyzed the data, prepared figures and/or tables, authored or reviewed drafts of the article, and approved the final draft.
- Jin Yan conceived and designed the experiments, performed the experiments, analyzed the data, prepared figures and/or tables, authored or reviewed drafts of the article, and approved the final draft.
- Guijiao Wang conceived and designed the experiments, performed the experiments, analyzed the data, prepared figures and/or tables, authored or reviewed drafts of the article, and approved the final draft.
- Yongfu Zhang conceived and designed the experiments, performed the experiments, analyzed the data, prepared figures and/or tables, authored or reviewed drafts of the article, and approved the final draft.

## Data Availability

The raw measurements are available in the Supplemental File.

## Supplemental Information

Supplemental information for this article can be found online at http://dx.doi.org/10.7717/peerj.19285#supplemental-information.

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
