# Peer review of "Enhancing effect of 5-azacytidine on saline–alkaline resistance of Akebia trifoliata and underlying physiological and transcriptomic mechanisms"

_PeerJ, doi:10.7717/peerj.19285_

## Round 0.1 · original submission · Major Revisions

Dear authors, I kindly ask you to make significant additions to the manuscript in accordance with the reviewers' comments. I hope that your detailed responses to each of the comments will allow the reviewers to approve this publication.

·

Basic reporting

This manuscript presents an interesting and comprehensive study on the effects of 5 - AzaC on the saline - alkaline resistance of Akebia trifoliata. The experimental design, which includes a wide range of physiological, biochemical, and transcriptomic analyses, is well - structured and provides valuable insights into the potential mechanisms by which 5 - AzaC can enhance the plant’s tolerance to saline - alkaline stress. However, there are many issues with the manuscript that require revision, as commented on below:
1. The introduction would be richer if it ended with some appropriate allusions to the uniqueness or innovativeness of the study, such as a reference to the gaps in current research in related fields and how this study fills those gaps.
2. How were the concentrations of salt and 5-AzaC used in the experiment determined? Were any pre-tests performed?
3. Scale should be added to drawings A, B of Figure 1.
4. “P < 0.05” should be replaced with “P < 0.05”. Please check the full manuscript and make corrections.
5. All genes appearing at the beginning of section 3.6 should be written in italics.
6. The authors sequenced the transcriptome of A. trifoliata under saline-alkaline stress and under 5-AzaC treatment. However, the GEO or SRA number of your RNA-seq information is not found. Moreover, the relevant transcriptome results (such as the logFoldChange value) should be shown with your qPCR results.

Experimental design

The experimental design of this study is basically reasonable.

Validity of the findings

Please see my "Basic reporting".

Additional comments

I have no any additional comments.

Reviewer 2 ·

Basic reporting

The manuscript “Enhancing effect of 5-azacytidine on saline-alkaline resistance of Akebia trifoliata and underlying physiological and transcriptomic mechanisms” features innovative and targeted material selection, and the experimental results are reliable. This study explores the potential role of 5-azacytidine in improving the saline-alkaline stress resistance of Akebia trifoliata and provides an in-depth analysis of its physiological and molecular mechanisms, which holds significant scientific value for understanding the saline-alkaline resistance mechanisms of Akebia trifoliata.

Experimental design

1. Was a Con + 5-AzaC treatment designed in the experiment? What is the effect of 5-azacytidine treatment on Akebia trifoliata in the absence of stress?
2. How were the concentrations of saline-alkaline stress and the application of 5-AzaC determined?
3. Provide the rationale for the selection of the 200 μmol/L 5-AzaC application concentration, including literature support or results from preliminary experiments.
4. Could the application of 5-azacytidine be described in more detail? For example, the application should be thorough enough to wet the entire leaf without forming droplets that run off.
5. The details of sampling in this experiment are not clearly described, and there are no replicate experiments. It is recommended to revise and supplement, and explain the reasons for the lack of replication.
6. Is the plant shown in Figure 1A not displayed completely? It is recommended to show the entire plant.
7. The discussion section needs further refinement, such as discussing the KEGG-enriched pathways together with physiological indicators and anatomical structures to explore their potential molecular mechanisms in more detail.
8. In the discussion section, compare the research results with existing literature in more depth, especially regarding the role of 5-AzaC in other plants.
9. The authors should provide more transcriptomic sequencing data, such as listing all differentially expressed genes (DEGs) and their expression values.

Validity of the findings

no comment

·

Basic reporting

The article is highly interesting, written with a high technical standard, and presents relevant findings for research on saline-alkaline stress tolerance in Akebia trifoliata. The results regarding the positive effect of 5-azacytidine (5-AzaC) treatment on resistance to this type of stress, through mechanisms such as enhanced antioxidant activity, regulation of key metabolic pathways, and reduction of oxidative damage, offer significant contributions to the field. Moreover, the study combines physiological, anatomical, and transcriptomic analyses in an integrated manner, providing a robust framework for the practical application of these findings in the cultivation of plants in saline-alkaline soils.
While the language used throughout the article is generally clear and technical, certain sections in the results may be difficult to follow due to their density. Additionally, repetitive sentences were identified within some sections, which affect the text's fluency. It is recommended to revise these areas to improve clarity and avoid redundancy. Specific details regarding these observations are provided in the attached PDF.
Regarding the results, it is recommended to include the full names of the genes at least the first time they are mentioned in the main text, as currently, only their abbreviations are provided in the figure legends. This would facilitate comprehension for non-specialized readers. Furthermore, inconsistency was observed in the way genes are referenced; some instances use abbreviations, while others use enzymatic codes. Standardizing this nomenclature would enhance clarity. Throughout the text, the transcriptomic analysis refers to differentially expressed genes (DEGs) as "DEG." It is recommended to consistently refer to them as "genes" or “transcripts” throughout the manuscript to enhance clarity and readability. Lastly, the discussion section contains extensive repetition of results, particularly concerning anatomical changes. Additionally, the discussion could be enriched by comparing the findings with previous studies on the effects of 5-AzaC on saline stress in other plants.
Regarding figures and tables, some figures are difficult to interpret due to their layout, and it is recommended to reorganize them for better clarity (e.g., Figure 3). Additionally, modifications to certain aspects of the figures are suggested to improve their comprehensibility. It is also recommended to revise the table to include the percentage of mapped reads and remove less informative columns, such as the number of bases and Q20 values.
More detailed information, including the specific locations of the observations mentioned, is provided in the attached PDF.

Experimental design

The study presents an original and highly relevant approach to understanding the effects of 5-azacytidine (5-AzaC) on Akebia trifoliata under saline-alkaline stress. While there is existing literature exploring the mechanisms by which this species copes with saline stress, this work addresses a significant gap by investigating the influence of this specific compound, with the combination of physiological, anatomical, and transcriptomic analyses, offering a valuable contribution to the field. The originality and focus of this study make it a pivotal step toward expanding our understanding of how A. trifoliata can adapt to challenging environmental conditions.
The study demonstrates a high level of technical rigor and provides a solid foundation for addressing the research question posed. The Materials and Methods section is well-written and includes detailed descriptions of the experiments, ensuring reproducibility. However, the experimental design could be further strengthened by the inclusion of an additional control. While a control with water was included, it is recommended to also incorporate a control with 5-azacytidine (5-AzaC) alone. As an inhibitor of DNA methylation, 5-AzaC might influence critical metabolic processes in the plant independently of saline-alkaline stress. Including such a control would help isolate the specific effects of 5-AzaC on plant physiology and provide a more nuanced understanding of its role in mitigating stress.
Additionally, while the statistical methods employed are robust, there is no mention of whether normality and homoscedasticity of the data were assessed prior to analysis. These checks are crucial to ensure the validity of parametric tests. Although an ANOVA is briefly mentioned in a figure legend, this information should be explicitly detailed in the Materials and Methods section to maintain methodological transparency and clarity. By addressing these aspects, the study’s findings would be further reinforced, offering even greater confidence in its conclusions.

Validity of the findings

This study addresses a key knowledge gap by investigating the impact of 5-azacytidine (5-AzaC) on Akebia trifoliata under saline-alkaline stress. The findings enhance our understanding of how this species copes with environmental stress and suggest potential agricultural applications, particularly for crops grown in difficult soil conditions. The results are solid and pave the way for further research, making this work a valuable contribution to the field. Additionally, replicating these findings in other species or under different environmental scenarios could provide further insights and strengthen the overall impact.
The data presented in the article are comprehensive and provide strong support for the conclusions drawn. However, it is recommended that the authors deposit the raw data, including transcriptomic and physiological datasets, in publicly available repositories. Doing so would not only enhance the reproducibility of the study but also facilitate its use by other researchers in related fields. Additionally, the statistical analyses are well-conducted, but as previously noted, detailing the steps taken to evaluate data normality and homoscedasticity would further strengthen the robustness of the results.
The conclusions are well-articulated and directly address the original research question, remaining firmly rooted in the data presented. The authors have successfully linked their findings to the broader context of stress tolerance in plants. The discussion effectively synthesizes the results and highlights their implications, making the study both informative and impactful.

Additional comments

I would like to express my sincere congratulations on the quality of this manuscript. The research presented is both innovative and highly significant, offering valuable insights into the role of 5-azacytidine in enhancing stress tolerance in *Akebia trifoliata*. The comprehensive experimental design, coupled with a thorough integration of physiological, anatomical, and transcriptomic analyses, provides a robust and well-structured framework for advancing our understanding in this field. This study addresses a critical knowledge gap and holds substantial potential for practical applications, particularly in improving crop resilience under saline-alkaline stress conditions. The findings make a significant contribution to the field and hold great promise for future research in this area. I strongly encourage the incorporation of the suggested revisions, as they would further enhance the clarity and impact of the work, ensuring its broader accessibility and relevance to the scientific community.

---

## Round 0.2 · Minor Revisions

Dear authors, the Discussion section should be changed radically, it should not repeat the results. The clarity and understandability of the article text should be improved. All the comments of the reviewers should be corrected. I hope that you will send me the final version of the article, which can be published.

·

Basic reporting

The authors have carefully revised the manuscript according to my previous comment.

Experimental design

It's OK.

Validity of the findings

It's OK.

Additional comments

There are still many formatting issues in the revised manuscript.

For example, the “t” in “table X" should be capitalized. The unit “day” should be abbreviated as “d”.

Line 132: "2.4 Assessment of changes in the anatomical structure of root, stem, and leaves of A. trifoliata": “A.” should not be abbreviated, must be the full name. The same goes for other titles, and please do not abbreviate in the title.

Line 168: "software and Microsoft Excel"->"and Microsoft Excel softwares" ;

Line 290: "GH3 auxin-responsive promoter (GH3)" ->"auxin-responsive promoter (GH3)" ;

·

Basic reporting

Several italicizations in the discussion section require review. In section 2.6, "Data Processing," it is necessary to include the statistical analysis conducted on the plants at the physiological, biochemical, morphological, and physiological levels. Additionally, in Figure 3, the legend needs to incorporate explanations for figures J, K, and L. The text's readability is somewhat compromised, and there is inconsistency in the use of abbreviations (gene acronyms or enzyme codes). Furthermore, at some points, the text is difficult to follow, and the discussion section repeats the results, which impacts the overall clarity and coherence of the article.

Experimental design

The idea of omitting a Con + 5-AzaC treatment group in the experiment poses significant concerns regarding the validity of the findings. Without this control, it becomes challenging to discern whether the observed effects are due to 5-AzaC's influence on salt stress mitigation or merely its inherent impact on plant physiology. By not considering the direct effects of 5-AzaC on plants, the experiment lacks a crucial comparison point, which undermines the reliability of the conclusions.

Validity of the findings

No comment

---

## Round 0.3 · Minor Revisions

Dear authors,

Your manuscript is almost ready to be accepted but some errors in the figure captions should be corrected. Figure 1: "Duncan's new complex range test" -- there is no such statistical test. The correct name is "Duncan's Multiple Range Test" (or sometimes referred to as "Duncan's New Multiple Range Test"); Figure 3: There are at least 6 grammatical errors in this caption. "was shows" should say "shows" (3 instances); "showed that" should say "shows" (3 instances).

·

Basic reporting

Thank you to the authors for their careful revisions. I think the manuscript can be accepted for publication.

Experimental design

It's OK.

Validity of the findings

It's OK.

Additional comments

No.

·

Basic reporting

No comments

Experimental design

No comments

Validity of the findings

No comments

Additional comments

No comments

---

## Round 0.4 · accepted · Accept

Dear authors, I am pleased to inform you that your manuscript has been accepted for publication in our journal.